# Natural Products and Obesity: A Focus on the Regulation of Mitotic Clonal Expansion during Adipogenesis

**DOI:** 10.3390/molecules24061157

**Published:** 2019-03-23

**Authors:** Eugene Chang, Choon Young Kim

**Affiliations:** 1Department of Nutritional Science and Food Management, Ewha Womans University, Seoul 03760, Korea; eugenics77@hotmail.com; 2Department of Food and Nutrition, Yeungnam University, Gyeongsan 38541, Gyeongbuk, Korea

**Keywords:** obesity, adipocyte differentiation, adipogenesis, natural products, 3T3-L1 preadipocytes

## Abstract

Obesity is recognized as a worldwide health crisis. Obesity and its associated health complications such as diabetes, dyslipidemia, hypertension, and cardiovascular diseases impose a big social and economic burden. In an effort to identify safe, efficient, and long-term effective methods to treat obesity, various natural products with potential for inhibiting adipogenesis were revealed. This review aimed to discuss the molecular mechanisms underlying adipogenesis and the inhibitory effects of various phytochemicals, including those from natural sources, on the early stage of adipogenesis. We discuss key steps (proliferation and cell cycle) and their regulators (cell-cycle regulator, transcription factors, and intracellular signaling pathways) at the early stage of adipocyte differentiation as the mechanisms responsible for obesity.

## 1. Introduction

Obesity is accepted as one of the great public health threats due to its close association with metabolic syndromes including type 2 diabetes, dyslipidemia, hypertension, and cardiovascular diseases [1,2]. In 2003, approximately 75 billion dollars was estimated as the annual extra medical cost due to obesity and obesity-related diseases, and this annual medical cost is estimated to increase by 48–66 billion dollars with a 130% increase in severe obesity prevalence by 2030 in the United States of America (USA) [3,4]. Given both the risk of obesity-related health complications and the economic burden, reducing body weight is regarded as a major health benefit [5]. Therapeutic strategies to treat and/or prevent obesity include lifestyle or behavior modification, nutrition education, a regular exercise program, and medication or surgery in the case of morbid obesity (body mass index (BMI) ≥ 40 kg/m^2^ or BMI ≥ 35 kg/m^2^ with comorbidities) [6]. Contrary to the expectation, it is estimated that more than 90% of the people who lose weight by lifestyle modification, such as dieting, return to their original weight within two to five years [7]. Additionally, some therapeutic approaches, such as anti-obesity drugs and laparoscopic adjustable gastric banding (bariatric surgery), are often the treatment modalities in obese patients, but surgery is limited to severe obese patients. Thus, anti-obesity drugs might be a promising solution to obesity. However, the possible side effects or adverse drug reactions of anti-obesity drugs; for example, fenfluramine- and dexfenfluramine-related potential damage to the heart valves, sibutramine-associated heart attack and strokes, rimonabant-related psychiatric adverse effects, and Xenical-induced severe liver injury are considered as a public health concern [8]. Hence, the efficacy, safety, and long-term effects of anti-obesity drugs must be evaluated before these are sold on the market. The growing threat of obesity to global health and the undetermined efficacy, safety, and long-term effects of anti-obesity drugs encouraged the researchers to put more attention and efforts to finding an efficient and safe anti-obesity ingredient.

Based on a market research report, the global market value for nutraceuticals including foods, beverages, and supplements was approximately 117.3 billion dollars in 2007, and the global market size is expected to reach 285.0 billion dollars by 2021 based on a 2017 global nutraceutical industry report [9]. The global dietary supplement market is estimated to reach 123 billion dollars due to the development of natural supplement products for weight loss. Between 2015 and 2025, the annual growth rate of the market for weight loss products is expected to be 7.4% [10]. In the category of dietary supplements, numerous potential materials from natural sources, including (1) natural plants such as herbs, fruits, and vegetables, (2) functional fatty acids such as polyunsaturated fatty acids and conjugated fatty acids, and (3) other natural dietary compounds and their active ingredients, are used as anti-obesity products [11]. This review describes the current knowledge on some of the phytochemicals showing the most promising effects on targeting white adipose tissue (WAT) development and changes during the development of obesity.

## 2. Adipose Biology during the Development of Obesity

Adipocytes play an important role in the progression of obesity and its associated metabolic syndromes. Owing to a chronic energy imbalance between calories consumed and expended, excess energy is stored in adipocytes in the form of triglycerides, which is the main feature of obesity [12]. Aside from being energy reservoirs, adipocytes were identified as active endocrine cells that produce and secrete a wide range of proteins, called adipokines. Adipokines (adipocyte-secreted hormones and proteins) include leptin, adiponectin, resistin, tumor necrosis factor alpha (TNF-α), interleukin 6, and monocyte chemoattractant protein 1 [13]. Adipokines are implicated in multiple cellular regulations related to energy homeostasis, inflammation, and insulin signaling, which in turn reflect health outcomes [14]. Therefore, it is important to understand the molecular mechanisms of adipose tissue formation and alterations during the progression of obesity for the prevention and treatment of obesity.

The key characteristic of obesity, that is, enlarged adipocyte tissue mass, is dependent on hypertrophy of preexisting individual adipocytes and/or hyperplasia (adipogenesis) due to the formation of new adipocytes from precursor cells [15]. Adipocyte hypertrophy is an increase in size of mature adipocytes as a result of lipid filling in preexisting fat cells. In contrast, adipocyte hyperplasia is a key process in determining the number of adipocytes, which mainly occur during childhood and adolescence. The cell number stays constant even after weight loss in adult and, thus, weight loss is mainly a result of the reduction in adipocyte volume [16]. In this key pathway, preadipocytes become mature adipocytes, which in turn increase the mass of adipose tissue [17]. Therefore, strategies that regulate both the size and number of adipocytes might be considered as a possible therapeutic approach in treating obesity. In this review, the molecular mechanisms via which phytochemicals influence adipogenesis, especially the early stage of adipogenesis, are discussed.

## 3. Regulation of Adipogenesis

The molecular and cellular processes of adipogenesis were extensively characterized using the 3T3-L1 preadipocyte fibroblast clonal cell line. The murine 3T3-L1 preadipocyte cell line is widely used in obesity research as the differentiation program is well characterized in this model [18,19]. Adipogenesis of 3T3-L1 preadipocytes involves growth arrest, mitotic clonal expansion (MCE), and terminal differentiation [19,20,21].

As briefly shown in Figure 1, the differentiation of 3T3-L1 post-confluent preadipocytes is induced by hormonal stimulation with dexamethasone (DEX), isobutylmethylxanthine (IBMX), and insulin. During this early stage of differentiation, termed as MCE, an irreversible commitment to differentiation occurs [22]. Adipogenic cocktail-induced differentiation stimulates clonal expansion, which in turn doubles cell number [23]. Since irreversibly committed preadipocytes undergo one or two rounds of replication during the first two days of differentiation, the induction of apoptosis in post-confluent differentiating cells leads to fewer adipocytes. Therefore, maturing preadipocytes could be an important target for natural products in the prevention and/or treatment of obesity.

### 3.1. Arrest and Progression of the Cell Cycle and Its Associated Regulatory Proteins

As a prerequisite for early adipogenesis, the cell cycle and its regulation play a pivotal role in the completion of MCE. Over-confluent proliferating 3T3-L1 preadipocytes are required since cell growth is inhibited by physical contact with neighboring cells two days before differentiation (Day −2), as shown in Figure 1. Upon confluence, preadipocytes are arrested in the gap 1 (G1) phase of the cell cycle with elevated levels of cyclin-dependent kinase (CDK) inhibitory proteins (CIPs), p21^CIP^ and p27^KIP1^, and hyperphosphorylated tumor suppressor retinoblastoma (Rb) protein [24].

During the early phase of differentiation, cell-cycle-arrested cells reenter the cell cycle and undergo one or two rounds of the cell cycle, regarded as MCE [25]. The activation and assembly of cyclin D to CDK4 and CDK6 and cyclin E with CDK2, and the degradation of the CDK inhibitor occur. The cyclin D and the CDK4/CDK6 complex is a regulator of the early G1 phase of the cell cycle, while the cyclin E and CDK2 complex is critical for the G1 and synthesis (S) phase transition, all of which helps cell-cycle-arrested cells to reenter the cell cycle, allowing them to progress to the G1/S phase [26,27]. Binding of CDK to cyclin is required for its kinase activity and phosphorylation of the Rb protein, a regulator of the E2F transcription factor family. The phosphorylation state of Rb protein changes from the hyperphosphorylated form, activating cell-cycle progression during the early stage of differentiation, to the hypophosphorylated form, suppressing the cell cycle during terminal differentiation [28]. Entry of the S phase occurs about 14 h after treatment with the adipogenic cocktail, and the highest DNA synthesis is observed about 18 h after hormonal stimulation [24,29]. Therefore, blockade or delay of cell-cycle progression by inactivating cell-cycle regulators and upregulation of CDK inhibitors during MCE might be an efficient way to inhibit adipogenesis.

### 3.2. Cascade of Transcriptional Factors during Mitotic Clonal Expansion

In an undifferentiated status, preadipocytes maintain high levels of preadipocyte factor-1 (Pref-1), CCAAT/enhancer binding protein (C/EBP) homologous protein (CHOP), Krüppel-like factor (KLF), GATA transcription factor, and Wingless/INT-1 protein (Wnt) signaling, while their levels dramatically decrease upon the induction of adipogenesis [30,31]. A sustained level of either of these proteins halts the adipogenesis program and keeps cells at the preadipocyte stage. Pref-1, especially the soluble form that is proteolytically generated from membrane-bound Pref-1, is known to inhibit adipocyte differentiation [32]. In contrast, Pref-1-null mice were shown to promote adiposity [33]. CHOP, a member of the C/EBP family of transcription factors, is able to form a dominant negative heterodimer binding with C/EBPβ, and this binding prevents the transactivation ability of C/EBPβ, resulting in the inhibition of adipogenesis [34]. GATA2 and GATA3, zinc finger transcription factors, are highly expressed in WAT, especially in the preadipocyte-enriched stromal–vascular fraction. The constitutive expression of GATA2 and GATA3 in adipocytes blocks adipogenesis [35]. The ectopic expression of Wnt10b inhibits adipogenesis [36]. Based on its high expression and function, an increase in transcription factors related to preadipocytes might be a good strategy to inhibit adipose tissue development during the progression of obesity.

Hormonal stimulation also leads to an adipogenic transcriptional cascade. Fully growth-arrested cells are challenged with a standard adipogenic cocktail consisting of DEX, IBMX, and insulin in fetal bovine serum (FBS)-containing medium. DEX, a synthetic glucocorticoid, stimulates the glucocorticoid receptor pathway and upregulates C/EBPβ expression, but not C/EBPδ expression [37]. DEX is also known to degrade Pref-1 [38]. On the other hand, IBMX, a phosphodiesterase inhibitor, activates the cAMP-dependent protein kinase pathway and induces C/EBPδ expression [39]. Meanwhile, insulin acts through the insulin-like growth factor 1 (IGF-1) receptor [40]. During the early stage of differentiation, FBS is known to downregulate CHOP10, releasing C/EBPβ to activate its downstream adipogenic gene expression and, thus, FBS is required for a rapid and full adipogenic phenotype [41]. These members of the C/EBP family of transcription factors, as well as peroxisome proliferator-activated receptor γ (PPARγ), play key roles in adipogenesis. The transcription factors, PPARγ and C/EBPα, activate the expression of genes related with lipid metabolism and terminate MCE [42]. With the sequential induction of the transcriptional factor cascade, the first hallmark of adipogenesis involves changes in intracellular lipid accumulation and cell shape.

#### 3.2.1. CCAAT/Enhancer Binding Protein β (C/EBPβ)

As early transcription factors, C/EBPβ and C/EBPδ expression is induced immediately after stimulation by adipogenic hormonal cocktails. Dual phosphorylation of mitogen-activated protein kinase (MAPK) and glycogen synthase kinase 3β (GSK3β) leads to phosphorylation and localization of C/EBPβ to the nucleus [43,44,45]. DNA-binding activity of C/EBPβ is not acquired until cells transverse the G1/S checkpoint about 12–16 h after hormone stimulation [46]. Hyperphosphorylation of C/EBPβ causes a conformational change that allows dimerization of C/EBPβ, thereby facilitating DNA-binding activity [47]. The phosphorylation of transcriptional activation of C/EBPβ leads to its subsequent transcriptional activation of PPARγ and C/EBPα, critical for terminal differentiation during MCE and DNA binding [44,46,48]. The critical role of C/EBPβ in adipogenesis was proven by suppressing adipogenesis and MCE, when functional DNA-binding and transactivation domains for C/EBPβ were disrupted [49]. Indeed, decreased nuclear localization of C/EBPβ by ceramide inhibits C/EBPα and PPARγ gene expression, thereby suppressing adipogenesis [50]. From this point of view, the blockade of C/EBPβ might be one of the best potential targets for the prevention and/or treatment of obesity.

#### 3.2.2. CCAAT/Enhancer Binding Protein α (C/EBPα)

C/EBPα is critical during the terminal differentiation of adipocytes. The *C/EBPα* gene possesses C/EBP regulatory elements in proximal promoters, and its expression is induced by C/EBPβ. As subsequent transcriptional activation after C/EBPβ activation occurs, C/EBPα can activate the expression of numerous downstream target genes such as *PPARγ*, lipoprotein lipase (*LPL*), sterol regulatory element-binding protein 1 (*SREBP-1*), and adipocyte fatty-acid-binding protein 2 (*aP2*) [51]. A lack of lipid storage in WAT was reported in C/EBPα knockout mice [52]. There are two isoforms of C/EBPα, p42 and p30, and only p42 has antiproliferative activity. Withdrawal from the cell cycle during terminal differentiation is controlled by C/EBPα, together with PPARγ and hypophosphorylation of Rb [53]. C/EBPα is activated when MCE ceases and cells become terminally differentiated [46].

#### 3.2.3. Peroxisome Proliferator-Activated Receptor γ (PPARγ)

As a master regulator of adipogenic programming, PPARγ expression is initially regulated by C/EBPβ. The blockade of C/EBPβ activity subsequently results in the inhibition of PPARγ, thus suppressing adipogenesis [54]. As it is not only essential for adipogenesis but also for the maintenance of the differentiated state, PPARγ is indispensable [19]. Concerted with C/EBPα, PPARγ plays a critical role in the expression of enzymes involved in triglyceride synthesis, such as lipin1 and acyl-CoA diacylglycerol acyltransferase 1 (DGAT1), which promotes lipid accumulation in the adipocyte [55,56,57]. Thus, adipocyte-specific PPARγ deletion blocks high-fat diet-induced obesity in mice [58].

#### 3.2.4. Gene Markers for Terminal Differentiation

Terminal differentiation refers to the status of cells that are withdrawn from the cell cycle [59]. An elevated lipid synthesis and a spherical morphology of mature adipocytes filled with lipid droplets are observed at this time point. Cells have increased lipid metabolism due to enhanced expression and activity of lipogenic and lipase enzymes including fatty-acid synthase, SREBP-1, and LPL. The expression level of adipocyte lipid binding protein (aP2) is also high [51,60]. Additionally, adipokines such as leptin, adiponectin, and resistin are highly expressed and secreted by mature adipocytes [61]. During adipogenesis, C/EBP activity regulates not only lipogenic SREBP1c induction but also inflammatory adipokine TNF-α [62,63]. Moreover, C/EBPβ activates the TNF-α gene promoter, confirming its proinflammatory effect. Furthermore, C/EBPα and PPARγ cross-regulate each other through positive feedback loops and transactivate downstream target genes such as aP2, LPL, and SREBP-1 [64,65].

### 3.3. Signaling Pathways Involved in Adipogenesis

A number of signaling pathways were identified in adipogenesis. Adipogenic hyperplasia is generally associated with the activation of cell proliferative signaling pathways, such as the insulin- and IGF-1-activated phosphoinositide 3-kinase/protein kinase B (PI3K/AKT) and mitogen-activated protein kinase/extracellular signal-regulated kinase (MAPK/ERK) pathways [66,67]. In contrast, Wnt/β-catenin signaling was inhibited during the early stages of adipogenesis [26,68].

#### 3.3.1. Phosphoinositide 3-Kinase/Protein Kinase B (PI3K/AKT) Pathway

Upon administering adipogenic hormonal cocktails, both insulin and IGF-1-induced protein kinase cascades are activated and play a pivotal role in G0/G1 cell-cycle arrest in 3T3-L1 preadipocytes [26]. The PI3K/AKT pathway regulates cell-cycle progression by modulating cyclin D and p27^KIP1^ expression [69,70]. Additionally, AKT acts as a negative regulator of GSK3β, which controls cyclin D1 stability and inhibits phosphorylation of Rb involved in the G1 phase [71]. Activation of the AKT pathway in 3T3-L1 preadipocytes contributes to adipocyte differentiation [72], whereas inactivation of the PI3K/AKT pathway inhibits adipogenesis [73]. Moreover, insulin-induced AKT activation inactivates the forkhead transcription factor via phosphorylation, which is essential for the progression of adipogenesis [74]. Overexpression of activated forkhead box class O1 (FoxO1) in adipocyte progenitor cells inhibits the progression of clonal expansion via a complex mechanism, including the induction of the cell-cycle inhibitors p21, p27, and pRb and the C/EBP dimerization partner CHOP10 [74]. In contrast, knockdown of FoxO1 markedly suppresses adipocyte differentiation accompanied with the decrease of PPARγ and C/EBPα expression, implying that FoxO1 plays an essential role in adipocyte differentiation, especially at the very early stage of terminal adipocyte differentiation [75].

#### 3.3.2. Mitogen-Activated Protein Kinase/Extracellular Signal-Regulated Kinase (MAPK/ERK) Pathway

Another major downstream component of the regulatory pathway, the MAPK/ERK pathway consists of ERKs, c-Jun N-terminal kinases (JNKs), and p38 MAPK in the MAPK family. In relation to early transcription factors, MAPK phosphorylates C/EBPβ, followed by GSK3β phosphorylation [43,44]. In detail, C/EBPβ phosphorylation on Thr-188 occurs via MAPK in the G1 phase (2–12 h after induction of differentiation) and via CDK2 in the S phase (12–24 h after induction), followed by additional phosphorylation on Thr-179 and Ser-184 by GSK3β [43,44,45]. Phosphorylation of C/EBPβ on Thr-188 is a prerequisite for phosphorylation by GSK3, which contributes to the localization of C/EBPβ to the centromere [43,44,45,47]. In relation to C/EBPβ-induced cyclin A and CDK2, intracellular MAPK signaling pathways play a major role in the regulation of cell proliferation and differentiation [25]. Moreover, ERK activation was shown to be essential for the induction of MCE and adipogenesis [76,77] by the involvement of cell-cycle progression [78]. In this regard, disruption or inhibition of the MAPK/ERK signaling pathway during MCE might be a potential target for adipocyte differentiation, adipose tissue formation, and obesity.

#### 3.3.3. Wingless/INT-1 Protein (Wnt)/β-Catenin Signaling

During the early stages of adipogenesis, adipogenic hormonal stimuli increase the expression of the early transcription factors, C/EBPβ and δ, together with the concurrent suppression of Wnt/β-catenin signaling [26,68]. Upon hormonal stimuli, increased expression of axin, GSK3β, and casein kinase 1 (CK1) leads to the inhibition of Wnt signaling. In the absence of Wnt signaling, β-catenin is located in the cytoplasm, phosphorylated, and degraded with ubiquitin mediation by a heteromeric multiprotein complex including axin, GSK3β, and CK1 [79,80,81]. The canonical Wnt signaling targets, β-catenin and Wnt receptor Fzd2, and the coreceptors Lrp5/Lrp6 are mostly expressed in preadipocytes, but their expression in adipocytes is limited [36,68,82,83]. Lrp6 initiates canonical Wnt signaling by promoting β-catenin stabilization, which causes the downstream inhibition of C/EBPβ and PPARγ [84,85,86]. In the nucleus, unphosphorylated β-catenin forms a transcription complex upon binding to transcription factor 4 (TCF4), which regulates downstream target genes of β-catenin, including *cyclin D1* and *c-Myc*, responsible for inhibition in C/EBPα or PPARγ-mediated adipogenesis [87,88,89,90]. Therefore, the stabilization or localization of β-catenin might be an important step for adipogenesis, which in turn may be a potential therapeutic strategy for the prevention and/or treatment of obesity and its associated metabolic diseases.

#### 3.3.4. AMP-Activated Protein Kinase (AMPK) Pathway

As a central regulator of cellular energy sensors, AMPK is involved in many cell functions including metabolic and biosynthetic pathways, organelle biogenesis, cell proliferation, and differentiation [91]. In human precursor cells, AMPK induces G1 cell-cycle arrest associated with decreased levels of cyclin D1 and hypophosphorylated pRb [92]. AMPK activation inhibits preadipocyte differentiation by suppressing transcription factors such as PPARγ, C/EBPα, and SREBP-1 required for adipogenesis [93,94]. One plausible explanation might be the involvement of AMPK-inducted sirtuin 1 (SIRT1) activation. As an nicotinamide adenine dinucleotide (NAD)^+^-dependent protein deacetylase, SIRT1 can promote lipolysis in 3T3-L1 adipocytes by repressing the activity of PPARγ2 [95]. Furthermore, AMPK increases fatty-acid β-oxidation via the inactivation of acetyl-CoA carboxylase and the increase of carnitine palmitoyltransferase 1 expression [96]. Therefore, AMPK is considered to be a target for the treatment of obesity.

## 4. Regulation of Adipogenesis by Various Phytochemicals

Numerous studies were conducted to investigate the effects of phytochemicals on adipocyte proliferation, formation, and alteration during the development of obesity. As mentioned above, one of the many possible potential targets for the prevention and/or treatment of obesity might be at least partially involved in the inhibition of the early stage of adipogenesis. Therefore, we discuss the influence of each natural component on cell proliferation, cell-growth arrest, cascade of transcriptional factor initiation, and intracellular signaling.

Table 1 includes bioactive components in the families of alkaloids, anthocyanins, asteraceae, coumarins, coumestans, curcuminoids, flavonols, flavones, flavonones, glucosinolates, isoflavonoids, isothiocyanates, phenolic acids, secoiridoids, stilbenes, and tannins, which inhibit adipocyte differentiation (adipogenesis) as evidenced by decreased oil Red O staining, triglyceride content, or glycerol-3-phosphate dehydrogenase assays, illustrating the decrement of adipocyte lipid accumulation. In this review, the molecular mechanisms via which phytochemicals listed in Table 1 influence obesity are discussed, focusing on the disruption, arrest, or delay of cell-cycle progression, the sequential cascade of transcriptional factors during MCE, and intracellular signaling during the early phase of adipocyte differentiation.

### 4.1. Cell-Cycle Arrest and Its Regulatory Proteins

Growth-arrested preadipocytes synchronously reenter the cell cycle and undergo one or two rounds of cell divisions, known as MCE [25]. Therefore, sustaining or increasing the cell population at one point of the cell cycle might be a good way to block adipocyte differentiation, thereby preventing/treating obesity. Given the association between cell-cycle arrest and adipogenesis, Table 1 shows the effects of phytochemicals on the cell cycle, cell-cycle regulators, and lipid accumulation. Delphinidin, a major anthocyanin widely found in pigmented fruits and vegetables [97]; apigenin, isolated from the flavonoid-rich fraction of *Daphne genkwa* Siebold et Zuccarini crude extracts [98]; sinigrin, a glucosinolate [99]; curcumin, derived from an Asian spice herb, *Curcuma longa*, and curcumin-modified forms, bisdemethyoxy-curcumin and curcumin-3,4-dichloro phenyl pyrazole [100,101,102]; dehydroleucodine, isolated from the aerial parts of *Artemisia douglasiana* [103]; sulforaphane, a naturally occurring isothiocyanate compound, produced in cruciferous vegetables such as broccoli and cabbage [67]; vitisin A, a resveratrol tetramer plentiful in the stembark of *Vitis* [104]; ellagic acid, present in raspberries, strawberries, walnuts, and pomegranate [105]; and oleuropein and hydroxytrosol, phenolic compounds [106] significantly inhibit intracellular lipid accumulation by increasing the cell population in the G0/G1 phase. Cell-cycle-arrested preadipocytes in the G1 phase of the cell cycle are associated with CDK inhibitory proteins, such as p21^CIP^ and p27^KIP1^, and Rb phosphorylation [24]. Indeed, delphinidin, dehydroleucodine, bisdemethyoxy-curcumin, curcumin-3,4-dichloro phenyl pyrazole, apigenin, sinigrin, sulforaphane, and vitisin A upregulate p21^CIP^ and/or p27^KIP1^ expression [67,97,98,99,102,103,104,107]. With the concomitant increase of cell number in the G0/G1 phase, decreased phosphorylation of Rb is observed in sulforaphane-, vitisin A-, or ellagic acid-treated cells [67,104,105]. In addition, fisetin, a flavonoid compound present in fruits and vegetables such as strawberries, persimmon, and onions, alters the expression profile of cell-cycle regulatory proteins including cyclin A, cyclin D1, CDK2, CDK4, and cyclin E, which are cell-cycle regulatory proteins in the G0/G1 phase [108].

Upon exposure to the adipogenic cocktail, the activation and assembly of cyclin D to CDK4 and CDK6 and of cyclin E with CDK2, and the degradation of the CDK inhibitor result in reentry into the cell cycle and G1/S phase progression [47,48]. With regard to a pivotal role of maturing preadipocytes in adipogenesis, curcumin derivative, curcumin-3,4-dichloro phenyl pyrazole [102], cocoa made from cacao (*Theobroma cacao* L.) [109], and caffeic acid phenethyl ester (CAPE), a polyphenol abundantly present in propolis [110], arrest the cell cycle at the G1–S checkpoint and decrease adipogenesis as featured by lipid accumulation. Moreover, cell entry into the S phase and the S to G2/mitosis (M) phase transition were blocked by curcumin, the yellow pigment and key bioactive compound found in the rhizome of the perennial herb turmeric [100,107]; epigallocatechin-3-gallate (EGCG), one of the main bioactive substances in tea [66,111]; genistein, an isoflavone present in soy [112]; piceatannol and resveratrol, naturally occurring stilbenoids found in grapes, sugarcane, berries, and peanuts, a natural stilbene [113,114]; and dieckol, containing primarily phlorotannins [115]. Furthermore, the cell-cycle regulatory machinery necessary for G1/S cell-cycle transition was investigated to underlie the inhibitory effects of caffeine or resveratrol on MCE. Caffeine (1,3,7-trimethylxanthine), a plant alkaloid found in coffee, chocolate, and tea, modulates cell-cycle progression through increased gene expression of p21^CIP^ and p27^KIP1^ and decreased CDK2 levels [116]. The 3T3-L1 cells treated with resveratrol exhibit reduced Rb phosphorylation and its associated ablation of cyclin A2 induction [117]. A line of evidence demonstrates at least partial modulation of the cell cycle and its associated regulatory or inhibitory proteins in phytochemical-mediated beneficial effects on obesity.

### 4.2. Cell Proliferation

Cell shape changes, existing lipid accumulation, and doubling cell number occur during the process of adipocyte differentiation [23]. A wide variety of studies showed that natural compounds inhibit adipocyte differentiation through the blockade of hormonal cocktail stimuli-induced cell proliferation, which may contribute their anti-obesity effects. Phytochemicals such as dehydroleucodine [103], caffeine [116], curcumin [100], EGCG [66,111], apigenin [98], rhamnetin [118], fisetin [108], genistein [112], sulforaphane [67], vitisin A [104], and resveratrol [114] remarkably suppressed adipogenic cocktail-induced proliferation. Decreasing post-confluent preadipocytes or inhibiting reentry to cell cycle is paid special attention to combat adipose tissue development upon chronic positive energy surplus. The induction of apoptosis in post-confluent differentiating cells contributes to lower adipogenesis. Therefore, cell apoptosis during the initiation of adipocyte differentiation could be an important target for natural products in the prevention of obesity.

### 4.3. Transcription Factors

Adipogenesis is accompanied by the expression of early adipogenic transcription factors, such as KLF4 and KLF5, C/EBPβ, C/EBPδ, and protein C-ets-2 (ETS2), whereas KLF2 expression is negatively regulated during MCE [19,26,31,119,120]. Messenger RNA (mRNA) expression of KLF2, a negative regulator of adipocyte differentiation, is suppressed by caffeine [116], whereas the early adipogenic transcription factors, KLF4 and KLF5, are upregulated by curcumin [100] and dieckol [115]. Similar to the decrement of KLF4 and KLF5 mRNA levels, dieckol also inhibits another early adipogenic transcription factor, ETS2 [115]. Berberine [121], apigenin [98], and genistein [122] remarkably increase C/EBP inhibitors, and CHOP10 is involved in the inhibition of adipocyte differentiation [74]. Like CHOP, basic helix–loop–helix homodimeric transcription repressors, differentiated embryo chondrocyte 1 (DEC1) and DEC2, are abundantly expressed in growth-arrested preadipocytes and are downregulated during the progression of adipocyte differentiation [41,123]. DEC1 and DEC2 inhibit the transcriptional activity of both C/EBPβ and C/EBPα required for adipogenic differentiation [123,124]. Berberine decreases adipogenesis by upregulating DEC2 mRNA levels [121].

C/EBPβ, an early adipogenic transcription factor, is thought to initiate MCE via DNA binding and phosphorylation of transcriptional activation, later coordinating the transcription complex network, leading to the formation of mature adipocytes via its subsequent transcriptional activation of PPARγ and C/EBPα [44,46,48]. Delphinidin, dehydroleucodine, curcumin, apigenin, isorhamnetin, piceatannol, and dieckol remarkably decrease an early adipogenic transcription factor, that is, C/EBPβ expression [97,98,99,103,113,115,125,126]. Additionally, decreased centromeric localization, DNA-binding activity, and the phosphorylation of C/EBPβ are found in apigenin-, sinigrin-, and genistein-administered adipocytes [98,99,122]. At the end of MCE, the involvement of PPARγ and C/EBPα in terminal differentiation is related to a rounded morphology with visible intracellular lipid droplets [19]. Phytochemicals listed in Table 1 show the inhibition of adipogenesis and the underlying mechanisms of their action associated with this suppression of transcription factors. Therefore, decreasing early adipogenic transcription markers and increasing transcription inhibitors related to adipogenesis, accompanied by inhibiting transcriptional cascades, contribute to the suppression of terminal adipogenic differentiation, which in turn prevents/treats adipose tissue development and obesity.

### 4.4. Intracellular Signaling Pathways

#### 4.4.1. Phosphoinositide 3-Kinase/Protein Kinase B (PI3K/AKT) Pathway

During the early stage of differentiation, hormonal cocktails activate PI3K/AKT and MAPK/ERK pathways [66,67]. The adipogenic hormonal cocktail stimulates the phosphorylation of insulin receptor (IR) and insulin receptor substrate 1, or IR kinase activity is inhibited in 3T3-L1 cells, which is reversed by cacao [109], resveratrol [114], piceatannol [113], or resveratrol [114]. In the same line, caffeine [116], dehydroleucodine [103], coumestrol [127], bisdemethoxycurcumin [107], curcumin-3,4-dichloro phenyl pyrazole [102], sulforaphane [67], cacao [109], CAPE [110], piceatannol [113], resveratrol [114,117], and dieckol [115] suppress adipogenic hormonal stimulation-induced AKT activation. Based on the role of the AKT pathway in adipocyte differentiation [72], these inhibitory effects of phytochemicals on AKT activation suggest that phytochemicals play a beneficial role in adipose formation and obesity progression. Additionally, the abundance and/or phosphorylation of GSK3β, a downstream marker of AKT, is altered by caffeine [116], delphinidin [97], coumestrol [127], and curcumin [128]. Other AKT downstream factors such as mammalian target of rapamycin (mTOR), p70S6 kinase, and ribosome protein S6 are remarkably inhibited by curcumin-3,4-dichloro phenyl pyrazole [102], fisetin [108], and cacao [109]. During the early stage of adipocyte differentiation, the potent inhibitory effects of active components on receptor tyrosine kinase, PI3K/AKT pathway, and AKT downstream factors, accompanied by the regulation of cell=cycle regulatory proteins and cell=cycle progression, suggest that phytochemicals might inhibit adipocyte differentiation and protect against obesity.

Given the close association between insulin-induced AKT activation and FoxO1 for the progression of adipogenesis at the very early stage of terminal adipocyte differentiation [74,75], FoxO1, a key modulator, might be a potential target. EGCG, a tea catechin, suppresses the clonal expansion of adipocytes by inactivating FoxO1 transcription via the PI3K/AKT and MEK/ERK pathways [66].

#### 4.4.2. Mitogen-Activated Protein Kinase/Extracellular Signal-Regulated Kinase (MAPK/ERK) Pathway

One of the major downstream phosphorylation cascades, MAPK/ERK signaling, influences the early phase of adipocyte differentiation via its associated phosphorylation of C/EBPβ and GSK3β, the centromere location of C/EBPβ, and cell-cycle progression [44,46,47,76,77]. Decreased phosphorylation of MAPKs was observed in adipocytes in the presence of curcumin [128], bisdemethoxycurcumin [107], sinigrin [99], sulforaphane [67], cocoa [109], CAPE [110], piceatannol [113], resveratrol [117], and dieckol [115]. The ability of natural active components to modulate the MAPK/ERK signaling pathway may explain, at least partly, the anti-adipogenic effects.

#### 4.4.3. Wingless/INT-1 Protein (Wnt)/β-Catenin Signaling

Accumulating evidence shows the inhibitory role of phytochemicals in adipogenesis through the induction of Wnt/β-catenin signaling. In preadipocytes, Wnt/β-catenin signaling, Wnt receptor Fzd2, and coreceptors Lrp5/Lrp6 are highly abundant, whereas their expression in adipocytes is decreased [36,68,82,83]. Dephosphorylation of β-catenin stimulates the expression of target genes, such as *cyclin D1*, *c-Myc*, and *c-Jun* [87,88]. Upon administering an adipogenic cocktail, GSK3β is phosphorylated, resulting in the suppression of Wnt signaling via β-catenin phosphorylation and ubiquitin-mediated degradation of β-catenin with axin, GSK3, and CK1 [79,80,81]. Therefore, the Wnt/β-catenin pathway might play an important role in early adipocyte differentiation.

As a natural product with a role in Wnt/β-catenin signaling pathway and adipogenesis, delphinidin results in a significant decrease of lipid accumulation in 3T3-L1 cells, accompanied by the activation of Wnt, Wnt receptor Fzd2, and the expression of the coreceptors Lrp5/Lrp6. Additionally, delphinidin treatment induces the stabilization of cytoplasmic β-catenin levels and its nuclear translocation with subsequent increased expression of its downstream target genes, *c-Myc* and *cyclin D1*. However, delphinidin suppresses GSK3β expression, a member of the β-catenin destruction complex [97]. Increased Lrp6 protein expression, resulting in the recovery of adipogenic cocktail-decreased β-catenin protein levels and its induced downregulation of Wnt10b, was observed in coumestrol-treated 3T3-L1 adipocytes. Furthermore, coumestrol upregulates mRNA and protein expressions of c-Myc and cyclin D1 [127]. Moreover, curcumin suppresses the expression of β-catenin destruction complex members, such as CK1, GSK3β, and axin, and upregulates the mRNA expression of Wnt10b, Fzd2, Lrp5/Lrp6, c-Myc, and cyclin D1 [128]. These results suggest that the Wnt signaling pathway might be involved in the inhibitory effects of delphinidin, coumestrol, and curcumin on adipogenesis.

#### 4.4.4. AMP-Activated Protein Kinase (AMPK) Pathway

AMPK activation suppresses the early adipogenic transcription markers including C/EBPα and PPARγ, which in turn inhibit preadipocyte differentiation [93,94]. Accompanied by the inhibition of transcription factors for adipogenesis, resveratrol [117] and dieckol [115] activated AMPK and, thus, decreased lipid accumulation in 3T3-L1 adipocytes. Thus, the inhibitory effect of resveratrol and dieckol on adipogenesis might be at least partially involved in AMPK activation.

## 5. Inhibition of Adipogenesis by Natural Product-Derived Bioactive Components

In nature, phytochemicals derived from natural products exist in a mixed form with other phytochemicals or nutrients such as dietary fiber, oils, vitamins, and minerals [135]. Accumulating evidence supports that plant extracts with multiple constituents exhibit superior biological function than isolated individual phytochemicals. Therefore, natural products execute synergistically favorable effects on health beyond the additive effects [136,137]. A mixture of individual phytochemicals with various phytochemicals or other components isolated from natural products might have a synergistic/additional inhibitory effect on adipogenesis.

### 5.1. Combination of Different Phytochemicals

Even though Table 1 shows that single phytochemicals could be able to inhibit adipogenesis, a group of phytochemical combinations might show synergistic effects on adipogenesis. For example, resveratrol treatment with quercetin synergistically suppressed adipogenesis in 3T3-L1 adipocytes, compared to the either resveratrol or quercetin alone group [138]. Interestingly, the combination of resveratrol, genistein, and quercetin demonstrated a superior inhibition of adipogenesis compared to that for each single phytochemical [139]. Moreover, phytochemical-rich extracts from natural product showed promising inhibitory effects on adipogenesis. Berry extracts from chokeberry *Aronia melanocarpa* (Michx.) Elliot, raspberry *Rubus idaeus* L., bilberry *Vaccinium myrtillus* L. and cranberry *Vaccinium macrocarpon* Aiton fruits rich in polyphenols such as anthocyanins, hydroxycinnamic acid derivatives, flavonols, and hydroxybenzoic acid derivatives decreased 3T3-L1 adipogenesis by downregulating adipogenic and lipogenic gene expression [140].

### 5.2. Combination of Phytochemicals with Other Compounds Isolated from Natural Product

Multiple constituents are found in natural products. The most extensively studied bioactive components are phytochemicals, but phytosterols, fatty acids, vitamins, and dietary fiber are also important bioactive compounds in natural products. Studies suggest that an individual phytochemical mixed with other bioactive components from natural products has synergistically reduced adipogenesis. Xanthohumol is a prenylated flavonoid found in the female flowers of the hops plant. When xanthohumol treatment was administered with honokiol (a lignin isolated from the bark, seed cone, and leaves of tress belonging to the genus *Magnolia*) [141] or with guggulsterone (a phytosterol isolated gum resin of the guggul plant) [142] in 3T3-L1 adipoctyes, adipogenesis was dramatically suppressed more than with xanthohumol alone. Even though a single treatment of guggulsterone or genistein slightly reduced adipogenesis, the combination of these compounds significantly lowered adipogenesis [138]. While genestein at 50 μM inhibited lipid accumulation by 40%, a genestain (50 μM) mixture with guggulsterone at 6.25 μM and 12.5 μM showed 83% and 94% inhibition of lipid accumulation, respectively. Similarly, genistein treatment with vitamin D exhibited a synergistic effect on the suppression of adipogenesis [143]. In vivo studies also suggested the synergistic activity of phytochemicals with other constituents in nature products. The administration of vitamin D combined with phytochemicals such as genestein, quercetin, and resveratrol synergistically inhibited body weight gain, together with improved bone mineral density in ovariectomized rats [101].

## 6. Conclusions

Obesity is an epidemic around the world. Multiple factors, such as dietary, lifestyle, genetic, environmental, and genetic factors, contribute to the development of obesity. For weight loss or appropriate weight maintenance, lifestyle and behavior modifications and maintaining a healthy lifestyle are required, although they are quite challenging. Therapeutic approaches and bariatric surgery are limited to severe obese patients. There are concerns about the use of anti-obesity drugs owing to fenfluramine-, dexfenfluramine-, and orlistat-induced severe adverse effects. Therefore, natural products and their phytochemicals are receiving more attention in order to prevent obesity, with a big expectation for their efficacy, safety, and long-term effects.

Obesity is characterized by enlarged adipocyte tissue mass resulting from an increased number and size of adipocytes. Decreasing proliferation and adipogenesis at the early stage of adipocyte differentiation might be potential target pathways for preventing or treating obesity. During the early stage of adipogenesis, dietary bioactive phytochemicals derived from natural products showed inhibitory effects on adipocyte lipid accumulation through the induction of apoptosis, cell-cycle arrest, transcription factors, and complex interconnected cell signaling pathways involved in regulating all the abovementioned processes. Therefore, it is anticipated that natural products could be potential functional ingredients in developing anti-obesity products.

## Figures and Tables

**Figure 1 molecules-24-01157-f001:**
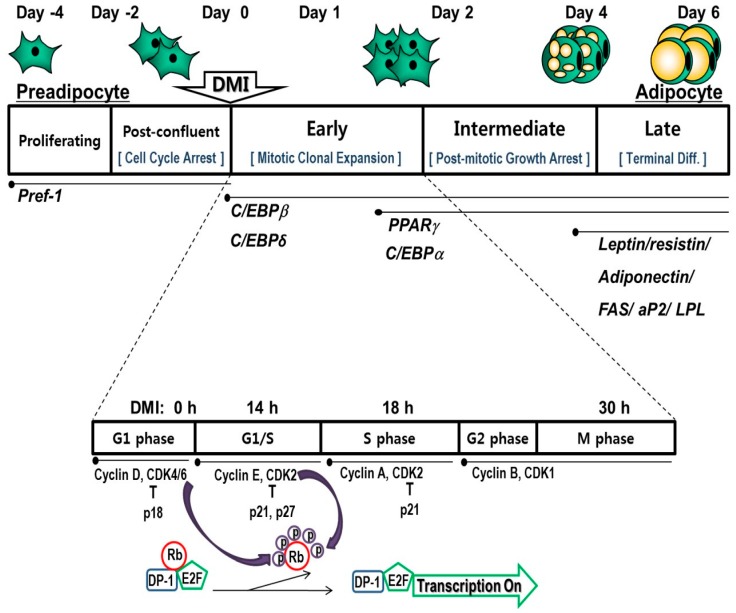
A scheme of chronological 3T3-L1 preadipocyte differentiation. Upon stimulating adipogenesis by a hormone cocktail, over-confluent preadipocytes undergo the distinct stages of differentiation: the early, intermediate, and late stages with changes in the expression of cell-cycle regulators and genes related to adipogenesis. aP2, adipocyte fatty-acid-binding protein 2; CDK, cyclin-dependent kinase; C/EBP, CCAAT/enhancer binding protein; DMI, adipogenic cocktail including dexamethasone (DEX), isobutylmethylxanthine (IBMX), and insulin; FAS, fatty-acid synthase; LPL, lipoprotein lipase; PPAR, peroxisome proliferator-activated receptor; Pref-1, preadipocyte factor-1; Rb, retinoblastoma.

**Table 1 molecules-24-01157-t001:** Effects of dietary natural products or their active components on changes during mitotic clonal expansion.

Family	Active Component	Tested Dose	Cell Proliferation	Cell Cycle	Cell-Cycle Regulators	Transcription Factor	Signaling Cascade	Intracellular Lipid Accumulation	Reference
**Alkaloids**	Berberine	0.625–10 μM				↓PPARγ, C/EBPα mRNA and protein levels		↓	[129]
	Berberine	1.5–12 μM				↓PPARγ, C/EBPα mRNA levels ↑CHOP, DEC2 mRNA levels		↓	[121]
	Caffeine	0.1–5 mM	↓		↑p21, p27 protein levels ↓CDK2 protein level	↓PPARγ, C/EBPα, C/EBPβ protein levels ↓C/EBPα, PPARα, FAS, aP2 mRNA levels ↑KLF2 mRNA level	↓ p-AKT, p-GSK3β protein levels = p-ERK protein level	↓	[116]
**Anthocyanins**	Delphinidin	10–150 μM		↑ G0/G1	↑p27 protein level ↓CDK2, CDK6 protein levels= CDK4, Cyclin D3 protein levels ↑cyclin D1 mRNA & protein levels	↓ C/EBPβ, C/EBPδ, C/EBPα mRNA levels ↓ PPARγ mRNA and protein levels	↑Wnt1, Wnt10b, Fzd2, and Lrp5 mRNA levels ↑β-catenin mRNA & protein levels, ↑β-catenin nuclear translocation ↑ mRNA & nuclear translocation of β-catenin ↓GSK3β mRNA & protein levels ↑c-Myc protein levels	↓	[97]
**Asteraceae**	Dehydroleu-codine	5–12.5 μM	↓	↑ G0/G1	↓CDK2, CDK4 protein level ↑p27 protein level	↓C/EBPβ, PPARγ mRNA levels	↓ p-AKT, p-Akt protein levels	↓	[103]
**Coumarins**	*p*-Coumaric acid	125–1000 μg/mL				↓PPARγ, C/EBPα protein levels	↑p-AMPK	↓	[130]
	Coumaric acid	0.1–0.2 mM				↓PPARγ2, C/EBPα mRNA and protein levels		↓	[131]
**Coumestans**	Coumestrol	20–60 μM				↓PPARγ, C/EBPα protein levels	↓ p-AKT, p-GSK3β ↑LRP6 protein level ↑β-catenin protein level ↑Wnt10b mRNA and protein levels ↑c-Myc, cyclin D1 mRNA and protein levels	↓	[127]
**Phenolic acid**	Curcumin	5–20 μM				↓ C/EBPα, PPARγ mNRA levels		↓	[132]
**Curcuminoids**	Curcumin	10–25 μM				↓ C/EBPα, PPARγ protein levels	↑Wnt10b, Fzd2, and Lrp5 mRNA levels ↑nuclear β-catenin protein ↓GSK-3β, CK1α, Axin protein levels ↑c-Myc, cyclin D1 mRNA and protein levels ↓p-ERK, p-JNK, p-p38 MAPK	↓	[128]
	Curcumin	5–30 μM				↓ C/EBPβ, PPARγ, C/EBPα mRNA levels		↓	[125]
	Curcumin	5–35 μM	↓	↓S and/or G2/M	↓Cyclin A, CDK2 protein levels	↓ KLF5, C/EBPβ, PPARγ, C/EBPα mRNA levels		↓	[100]
	Bisdemethoxycurcumin (BDMC)	5–25 μM		↑G0/G1	↓cyclin A, cyclin B protein levels ↑p21 protein level = CDK2 and 4, Cyclin D and E protein levels	↓PPARγ, C/EBPα protein levels	↓p-ERK1/2, p-JNK/ = p-p38 MAPK ↓p-Akt	↓	[107]
	Curcumin-3,4-dichloro phenyl pyrazole	5–20 uM		↑G1, S	↓CyclinD1, CyclinD3, CDK2, CDK4, CDK6 protein levels ↑p27 protein level	↓PPARγ2, C/EBPα mRNA and protein levels	= Wnt3a, GATA, β-catenin, p-AMPK protein levels ↓ p-AKT, mTOR protein levels	↓	[102]
**Flavanols**	Catechin 3 -gallate (CG)	5–30 μM				↓ C/EBPα, PPARγ protein levels		↓	[133]
	Epicatechin 3-gallate (ECG)	5–30 μM				↓ C/EBPα, PPARγ protein levels		↓	[133]
	Epigallocatechin-3-gallate (EGCG)	0.1–10 μM	↓	↑G2/M		↓ C/EBPα, PPARγ mRNA levels		↓	[111]
	Epigallocatechin-3-gallate (EGCG)	100 μM	↓	↓G0/G1 ↑S		↓ C/EBPα, PPARγ mRNA levels	↓ FoxO1 mRNA level	↓	[66]
**Flavones**	Apigenin	30–70 μM	↓	↑G0/G1	↓CyclinD1, CDK4 protein levels ↑p27 protein level	↓DNA-binding activity of C/EBPβ ↓C/EBPβ protein level ↑p- C/EBPβ, CHOP-10 protein levels		↓	[98]
**Flavonoids**	Isorhamnetin	1–50 μM				↓C/EBPβ, C/EBPδ mRNA levels ↓PPARγ, C/EBPα mRNA levels		↓	[126]
**Flavonoids**	Rhamnetin	10–40 μM	↓			↓PPARγ, C/EBPα mRNA and protein levels		↓	[118]
**Flavonoids**	Fisetin	10–30 μM	↓		↓cyclin A, cyclin D1, CDK4 protein levels ↑p27 protein level = Cyclin E, CDK2 protein levels	↓PPARγ protein level	= p-ERK and p-AKT ↓ p-S6 (p70S6K activity)	↓	[108]
**Glucosinolates**	Sinigrin (2-propenyl glucosinolate)	1–100 μg/mL		↑G0/G1	↑p21, p27 protein levels ↓CDK2 protein level	↓p-C/EBPβ ↓PPARγ, C/EBPα protein & mRNA levels	↓ p-ERK, p-JNK and p-p38 MAPK ↑p-AMPK	↓	[99]
**Flavonoids:** **Isoflavonoids**	Genistein	5–100 μM	-	-		↓PPARγ protein level		↓	[134]
	Genistein	100 μM				↓PPARγ, C/EBPα protein levels = C/EBPβ protein level ↓DNA-binding activity of C/EBPβ ↑CHOP protein level		↓	[122]
	Genistein	50 μM	↓	↑S	↑cyclin A protein level = p27, p21, cyclin E, CDK2 protein levels	↓Centromeric localization of C/EBPβ	= p-MAPK, GSK3β protein levels	↓	[112]
**Isothiocyanates**	Sulforaphane	5–20 μM	↓	↑G0/G1	↑ p27 protein level / ↓p-Rb ↓ cyclin D1, CDK2, CDK4, cyclin A protein levels	↓PPARγ, C/EBPα protein levels	↓p-AKT, p-ERK	↓	[67]
**Phenolic acid**	Cocoa	100–200 μg/mL		↑G1-S		↓PPARγ, C/EBPα mRNA and protein levels	↓p- ERK, p-AKT, mTOR, p70S6K protein levels ↓ Insulin receptor kinase	↓	[109]
**Phenolic acids**	Caffeic acid phenethyl ester	10–40 μM		↑G1/S	↓cyclin D1 mRNA and protein levels	↓PPARγ, C/EBPα protein levels	↓p-ERK, p-AKT	↓	[110]
**Secoiridoids**	Hydroxytyrosol	50–150 μM		↑G0/G1 ↑S		↓PPARγ, C/EBPα mRNA levels		↓	[106]
	Oleuropein	100–300 μM		↑G0/G1 ↑S		↓PPARγ, C/EBPα mRNA levels		↓	[106]
**Stilbenes**	Vitisin A	1–10 μM	↓	↑G0/G1	↓p-Rb ↑p21 protein level	↓PPARγ protein level and activity	= p-ERK, p-AKT	↓	[104]
	Ellagic acid	10–20 μM		↑G0/G1	↓p-Rb, cyclin A protein levels	↓C/EBPα protein level and DNA-binding activity ↓PPARγ mRNA and protein levels		↓	[105]
	Piceatannol	10–50 μM	NS	↓S and G2/M		↓PPARγ, C/EBPα protein and mRNA levels ↓C/EBPβ mRNA level	↓ p-IR, p-IRS-1, p-AKT, p-ERK1/2	↓	[113]
	Resveratrol	25–50 μM	↓	↑G1/S	↓ cyclin A, CDK2 protein levels	↓PPARγ, C/EBPα protein levels	↓p-AKT, p-IR ↓IR kinase activity	↓	[114]
	Resveratrol	20 μM			↓p-Rb, Cyclin A and D1, p21 protein levels	↓PPARγ, C/EBPα protein level ↓PPARγ activity	↓p-AKT, p-ERK ↑p-AMPK	↓	[117]
**Tannins**	Dieckol	25–100 μM	↓	↑G1↓S	↓ cyclin A and D, p-Rb, CDK2 protein levels ↑p27 protein level	↓C/EBPβ, C/EBPδ, KLF4, KLF5, ETS2 mRNA levels	↓p- ERK, p-AKT ↑p-AMPK	↓	[115]

AKT, protein kinase B; AMPK, AMP-activated protein kinase; aP2, adipocyte fatty-acid-binding protein 2; CDK, cyclin-dependent kinase; C/EBP, CCAAT/enhancer binding protein; CHOP, CCAAT/enhancer binding protein (C/EBP) homologous protein; DEC, differentiated embryo chondrocyte; ERK, extracellular signal-regulated kinase; ETS2, protein C-ets-2; FAS, fatty-acid synthase; FoxO1, forkhead box class O1; Fzd2, frizzled-2; GSK3β, glycogen synthase kinase 3β; IR, insulin receptor; IRS-1, insulin receptor substrate-1; JNK, c-Jun N-terminal kinases; KLF, Krüppel-like factor; Lrp5, lipoprotein receptor-related protein 5; NS, not significant; mTOR, mammalian target of rapamycin; p38 MAPK, P38 mitogen-activated protein kinase; PPAR, peroxisome proliferator-activated receptor; Rb, retinoblastoma; Wnt, wingless/int-1 protein; = not changed.

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
