# Peer review of "Natural Products and Obesity: A Focus on the Regulation of Mitotic Clonal Expansion during Adipogenesis"

_molecules, 2019, doi:10.3390/molecules24061157_

Round 1

Reviewer 1 Report

TO THE AUTHORS 

The authors have conducted an extensive search of the existing literature and have summarized some of the information related to the molecular mechanisms underlying adipogenesis and the inhibitory effects of various phytochemicals. Phytochemicals are classified into groups and the authors have listed the effects of these phytochemicals on some changes during mitotic clonal expansion. This review can be strengthened by addressing the following issues:

1)  It is opportune to include more recent references in some paragraphs. In the following sections, more recent references can be placed.

- In page # 1, first paragraph, for lines 24-26 and 28-34.

- In page # 13, lines 389-391

2) It is suggested to add to Table 1 the compounds and results found in the following   originals articles, in order to update the information submitted PMID: 29182628, PMID: 28732567, PMID: 29663555.

3) It is recommended to update the data in the line 47 in page # 2, the amount of 117.3 billion dollars today must be higher.

4) It is proposed for the enrichment of the conclusions to mention some severe adverse effects of anti-obesity drugs.

Minor Issues:

1)  Reference # 20. Correct: Gd – GD.

2)  In the paragraph of section 4.4.2 MAPK/ERK pathway line 5, please change CACAO by COCOA). The same change in table 1 on the reference 117.  

3)  Page # 1, line 315. Intracellular, capital letter.

Author Response

Thank you for the opportunity to respond to the Reviewer’s excellent suggestions for our manuscript titled, “Natural products and obesity: A focus on the regulation of mitotic clonal expansion during adipogenesis”. We have replied in detail to each suggestion in the attached response to reviewers. We hope that the manuscript may now meet the reviewers and editors’ expectation.

Comments of Reviewer 1

1)  It is opportune to include more recent references in some paragraphs. In the following sections, more recent references can be placed.

- In page # 1, first paragraph, for lines 24-26 and 28-34.

- In page # 13, lines 389-391

Author answer: We really appreciate and agree to reviewer’s valuable opinion. Based on the reviewer’s critical comment, more recent references were cited.

2) It is suggested to add to Table 1 the compounds and results found in the following   originals articles, in order to update the information submitted PMID: 29182628, PMID: 28732567, PMID: 29663555.

Author answer: The results of articles that reviewer suggested were added in the text and Table 1.

3) It is recommended to update the data in the line 47 in page # 2, the amount of 117.3 billion dollars today must be higher.

Author answer: Recent information related to the medical cost due to obesity was added.

4) It is proposed for the enrichment of the conclusions to mention some severe adverse effects of anti-obesity drugs.

Author answer: Side effects of anti-obesity drugs are explained in the introduction and conclusion sections.

Minor Issues:

1)  Reference # 20. Correct: Gd – GD.

Author answer: It has been corrected.

2)  In the paragraph of section 4.4.2 MAPK/ERK pathway line 5, please change CACAO by COCOA). The same change in table 1 on the reference 117.  

Author answer: It has been corrected.

3)  Page # 1, line 315. Intracellular, capital letter.

Author answer: It has been corrected.

Reviewer 2 Report

Eugene Chang and Choon Young Kim have summarized the literature related to the influence of natural products on the phase of clonal expansion during adipocyte differentiation. In the first part of the review, they describe adipocyte differentiation and the processes and pathways involved in it. Subsequently, they give an overview of natural products that affect adipogenesis by targeting the highlighted mechanisms and signaling pathways during the early phase of clonal expansion.

In general, this is a well written and balanced review article, which I enjoyed reading.

Suggestions for improvement:

The authors have summarized the cited literature in a comprehensive table. For better comparison of the effects of different compounds, it would be informative to add the concentrations that were tested in the respective studies.

Xanthohumol and Withaferin A are phytochemicals with a broad spectrum of activities in vitro and in vivo that were tested for their influence on adipogenesis (e.g. PMID 17874298, 19346589). They should be included.

There are numerous studies where combinations of compounds have been tested in the 3T3L1 model. Considering the fact that diet also represents a combination of compounds, it would be informative to include these studies, especially when combinations are more active than either compound tested alone. Examples of combination studies: PMID 17318368, 18369342,18239559,18433793,19735186,17959142,18525110, 177116704,19053873.

Currently, the review is based on in vitro (mechanistic) studies. Nevertheless, it would be important to (at least) exemplarily include studies, where the impact of the phytochemicals and natural compounds on adipogenesis or fat accumulation has been tested in vivo. This would emphasize the relevance of the cited studies for the in vivo or even human situation. Examples include Xanthohumol, genistein, Withaferin A etc.

In general, the list of references is a little outdated. There are few citations from 2016 to 2019.

The text should be checked for some grammatical errors. The references should be carefully checked with respect to hyphenation.  

Author Response

Reponses to reviewer

Thank you for the opportunity to respond to the Reviewer’s excellent suggestions for our manuscript titled, “Natural products and obesity: A focus on the regulation of mitotic clonal expansion during adipogenesis”. We have replied in detail to each suggestion in the attached response to reviewers. We hope that the manuscript may now meet the reviewers and editors’ expectation.

Comments of Reviewer 2

1) The authors have summarized the cited literature in a comprehensive table. For better comparison of the effects of different compounds, it would be informative to add the concentrations that were tested in the respective studies.

Author answer: The concentration of the compounds was added in Table 1.

2) Xanthohumol and are phytochemicals with a broad spectrum of activities in vitro and in vivo that were tested for their influence on adipogenesis (e.g. PMID 17874298, 19346589). They should be included.

Author answer: The results of articles that reviewer suggested were added in the text and Table 1.

3) There are numerous studies where combinations of compounds have been tested in the 3T3-L1 model. Considering the fact that diet also represents a combination of compounds, it would be informative to include these studies, especially when combinations are more active than either compound tested alone. Examples of combination studies: PMID 17318368, 18369342,18239559,18433793,19735186,17959142,18525110, 177116704,19053873.

Author answer: New sections with title of “Inhibition of adipogenesis by natural product-derived bioactive components” were added to describe the effect of combination of different phytochemicals and combination of phytochemicals with other compounds.

4) Currently, the review is based on in vitro (mechanistic) studies. Nevertheless, it would be important to (at least) exemplarily include studies, where the impact of the phytochemicals and natural compounds on adipogenesis or fat accumulation has been tested in vivo. This would emphasize the relevance of the cited studies for the in vivo or even human situation. Examples include Xanthohumol, genistein, Withaferin A etc.

Author answer: The results of articles that reviewer suggested were added.

5) In general, the list of references is a little outdated. There are few citations from 2016 to 2019.

Author answer: Based on the reviewer’s critical comment, more recent references were cited.

6) The text should be checked for some grammatical errors.

Author answer: Manuscript was reviewed by expert editors through English editing service.

7) The references should be carefully checked with respect to hyphenation.  

Author answer: The format of reference was carefully checked.
